# Distributed task-specific processing of somatosensory feedback for voluntary motor control

Mohsen Omrani[1,2], Chantelle D Murnaghan[1], J Andrew Pruszynski[1,3], Stephen H Scott[1,4,5]*

[1]Centre for Neuroscience Studies, Queen's Univertsity, Kingston, Canada; [2]Brain Health Institute, Rutgers Biomedical and Health Sciences, New Jersey, United States; [3]Physiology and Pharmacology, Schulich School of Medicine and Dentistry, Robarts Research Institute, University of Western Ontario, Ontario, Canada; [4]Department of Biomedical and Molecular Sciences, Queen's University, Kingston, Canada; [5]Department of Medicine, Queen's University, Kingston, Canada

**Abstract** Corrective responses to limb disturbances are surprisingly complex, but the neural basis of these goal-directed responses is poorly understood. Here we show that somatosensory feedback is transmitted to many sensory and motor cortical regions within 25 ms of a mechanical disturbance applied to the monkey's arm. When limb feedback was salient to an ongoing motor action (task engagement), neurons in parietal area 5 immediately (~25 ms) increased their response to limb disturbances, whereas neurons in other regions did not alter their response until 15 to 40 ms later. In contrast, initiation of a motor action elicited by a limb disturbance (target selection) altered neural responses in primary motor cortex ~65 ms after the limb disturbance, and then in dorsal premotor cortex, with no effect in parietal regions until 150 ms post-perturbation. Our findings highlight broad parietofrontal circuits that provide the neural substrate for goal-directed corrections, an essential aspect of highly skilled motor behaviors.

*For correspondence: steve.scott@queensu.ca

## Introduction

The motor system is capable of performing a wide range of skilled motor behaviors, from pouring tea in a cup to catching a ball while running. Optimal feedback control (OFC) has become an influential theory for interpreting voluntary motor control (*Todorov and Jordan, 2002*; *Scott, 2004*; *2012*). OFC includes state estimation (i.e. position and velocity of the body segments) based on sensory and internally generated feedback. It also includes a control policy that uses these state estimates to generate motor commands to muscles that generate movement to attain a behavioral goal. Importantly, feedback gains within the control policy are selected based on the behavioral goal.

As feedback is an essential component of OFC, an important approach to probe the properties of the motor system is to use small mechanical loads applied to the limb to disturb the motor system and observe how it responds to attain different behavioral goals (*Scott, 2004*). This approach has been used to show how muscle responses are modified in under 100ms due to behavioral factors such as task instruction (*Hammond et al., 1956*; *Rothwell et al., 1980*; *Pruszynski et al., 2008*; *Shemmell et al., 2009*), the properties of the spatial target and selection of alternate goals (*Yang et al., 2011*; *Nashed et al., 2012*; *Selen et al., 2012*), location of obstacles in the environment (*Nashed et al., 2012*; *2014*), mechanical properties of the limb and environment (*Kurtzer et al., 2008*; *Shemmell et al., 2010*; *Cluff and Scott, 2013*; *Weiler et al., 2015*), and

**eLife digest** Humans and other animals can change a movement they are making in a split second, such as when a basketball player has to move around an unexpected opponent to shoot a ball through the hoop. These on-the-fly corrections rely on information about the movement that comes in from the senses. However, it was unclear how the brain and spinal cord process this sensory information to guide movement.

Omrani et al. have now recorded electrical activity from the brains of monkeys while the animals tried to keep their hand at a target. Each monkey wore a robotic exoskeleton that would occasionally move the monkey's arm. Even if the monkey was not engaged in a motor task, a small nudge of the limb by the robot caused neural activity to spread rapidly throughout the sensory and motor regions of the cerebral cortex (the outer layer of the brain).

In some trials, when the monkey was actively trying to keep its hand at a target, the robot would again nudge the monkey's arm slightly. Omrani et al. observed that within 25 milliseconds of this nudge, the activity in an area of the cortex called parietal area 5 responded even more, suggesting that this area was using information from the senses to actively deal with the change in arm position. This change in activity then spread to other parts of the brain.

In another set of trials, the monkey was trained to move to a second target if the robot nudged its arm. In this case, the activity in an area called the primary motor cortex increased even more, likely supporting the monkey's ability to rapidly move to this second target. Overall, the study by Omrani et al. highlights the complex way that sensory feedback is processed in the cerebral cortex, supporting our ability to perform highly skilled motor actions.

timing constraints or task urgency (*Crevecoeur et al., 2013*; *Omrani et al., 2013*; *Cluff and Scott, 2015*).

The neural basis of this task-dependent feedback processing is, however, largely unexplored. Frontoparietal circuits are known to play an important role in voluntary control, but the focus for the last 30 years has been on motor planning and the initiation of motor actions (*Kalaska et al., 1997*; *Andersen and Cui, 2009*). While the mathematical details of OFC are not implemented at the neural level, the OFC framework creates an important dichotomy between state estimation and the control policy. From this perspective, perturbation-related activity in a brain region that is not influenced by the behavioral goal would be associated with the former, whereas perturbation-related activity that is modified by the behavioral goal, would be associated with the latter.

There are sporadic observations that several sensory and motor cortical areas respond to mechanical loads applied to the limb (*Tanji et al., 1980*; *Chapman et al., 1984*; *Crammond et al., 1986*; *Boudreau et al., 2001*; *Weber and He, 2004*; *London and Miller, 2012*; *Spieser et al., 2013*). However, task-dependent changes in neural activity have only been examined in primary sensory (S1) and motor cortices (M1) (*Evarts and Tanji, 1974*; *Wolpaw, 1980*; *Omrani et al., 2014*; *Pruszynski et al., 2014*). Thus, the extent to which other cortical regions are involved in generating these task-dependent feedback responses (i.e. part of the control policy) remains largely unexplored. Here we use mechanical disturbances applied to the arm of non-human primates (NHPs) under multiple behavioral conditions to reveal task-dependent feedback processing across frontoparietal circuits.

## Results

### Rapid transmission of limb feedback across sensory and motor cortex

Our first experiment examined the timing and magnitude of neural responses in a number of frontal and parietal cortical regions elicited by mechanical loads applied to the forelimb as monkeys maintained their hands at a central target (Posture Task, *Figure 1A*, *Herter et al., 2007*). The activity of 611 neurons was recorded in 5 different cortical regions associated with voluntary motor control (A5:posterior parietal area 5, A2:primary somatosensory area 2, S1:primary somatosensory area 1 & 3, M1:primary motor cortex and PMd:dorsal premotor cortex, See Materials and methods). We

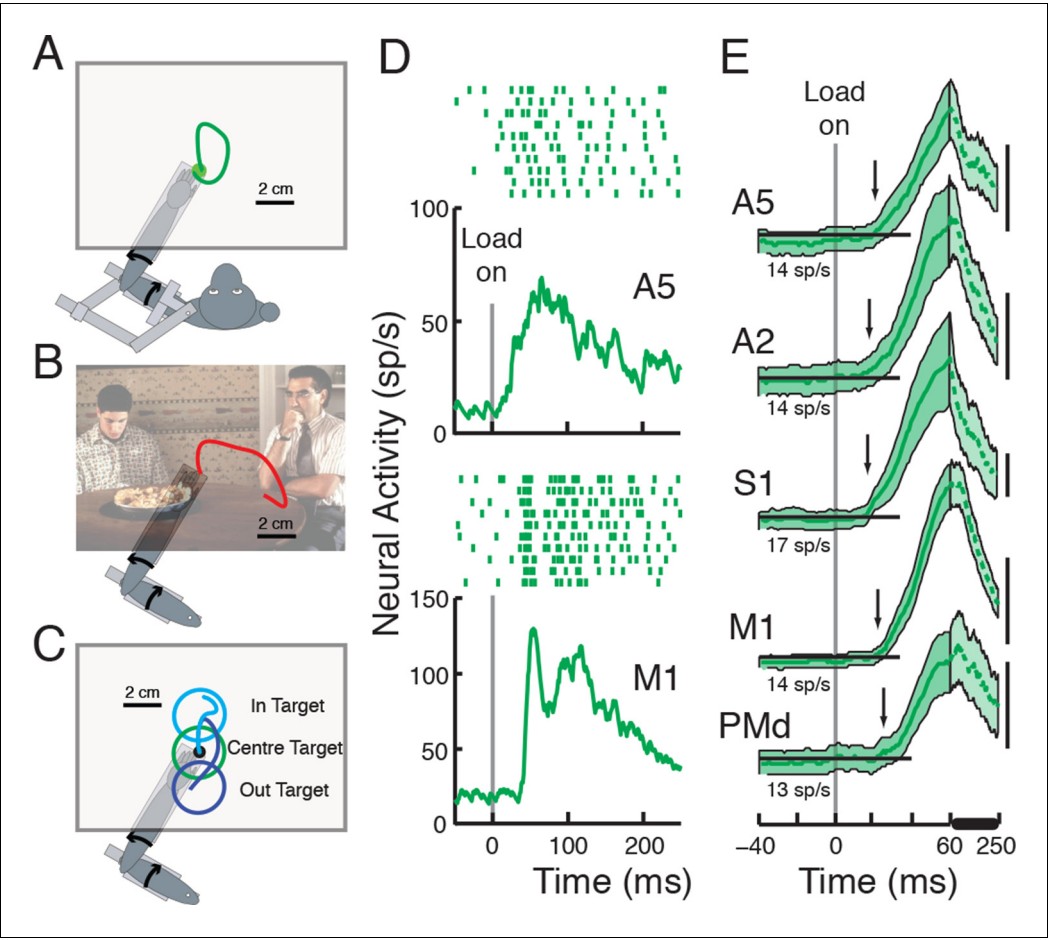

**Figure 1.** Behavioral tasks and perturbation responses across cortical areas. Each task varied in how the monkeys were instructed to respond to perturbations applied to the limb: correct for the disturbance in the Posture Task (**A**), not required to respond in the Movie Task (**B**), move to a spatial target in the IN/OUT Task (**C**). (**D**) Perturbation response, in the cell's preferred torque direction, in the Posture Task of a neuron in A5 and in M1. Tick marks denote action potentials, each row representing a separate trial. (**E**) Population responses in each cortical area (Mean±2SEM). Arrows depict when population signal surpassed threshold (baseline + 3SD) for >20 ms. Pre-perturbation baseline activity (-100:0 ms pre-perturbation, horizontal insets) in each area is stated below each corresponding population response. Scale bars denote 20 sp/s. Population activity between 60–250 ms (denoted by thick horizontal line) is compressed for visualization purposes. The 'American Pie' picture is reproduced with permission from Universal Studios.
Figure 1 part B photo (from the film American Pie) is reproduced with permission from Universal Studios (© copyright Universal Pictures, 1999, All Rights Reserved).
The following figure supplement is available for figure 1:

**Figure supplement 1.** Locations of recorded neurons from each session in Monkey P.

found many neurons in each cortical region displayed significant perturbation-related activity within 100 ms of loads being applied in their preferred torque direction (*Figure 1D*, refer to *Table 1* for details on the number of neurons recorded in each area and neurons responsive to perturbations).

Although neurons in each cortical region responded to the applied perturbations, the magnitude of the perturbation-related responses varied across cortical regions (one way ANOVA, p<0.0001, $F_{(df=4, error=439)}$=10.62, See *Table 2* for details). A post-hoc analysis revealed that the response in S1 was significantly larger than responses in all other areas (p<0.001, Tukey's least significant difference

**Table 1.** Number of neurons in each area recorded in each monkey and neurons with significant change in activity (two-sample t-test, p<0.05) in response to perturbation or across tasks.

|  | A5 | A2 | S1 | M1 | PMd |
|---|---|---|---|---|---|
| #cells in Monkey P | 70 | 0 | 40 | 147 | 50 |
| #cells in Monkey X | 0 | 0 | 0 | 90 | 0 |
| #cells in Monkey A | 55 | 55 | 0 | 35 | 69 |
| #cells with significant perturbation response (/recorded neurons) in the Posture Task | 87/125 | 47/55 | 33/40 | 213/272 | 64/119 |
| #cells with significant perturbation response recorded in both the Posture & Movie Tasks | 65/87 | 36/43 | 21/26 | 129/160 | 54/100 |
| #cells with significant change in activity between Posture & Movie Tasks (/significant perturbation response) | 22/65 | 4/36 | 6/21 | 74/129 | 17/54 |
| #cells with significant reduction in activity in Movie Task (/significant task effect) | 20/22 | 3/4 | 5/6 | 65/74 | 15/17 |
| #cells with significant perturbation response (/recorded neurons) in the IN-OUT Task | 35/41 | 21/25 | 18/18 | 83/92 | 58/87 |
| #cells with significant change in activity in the IN/OUT task (/significant perturbation response) | 9/35 | 0/21 | 2/18 | 31/83 | 16/58 |
| #cells with significant increase in activity in OUT target (/significant task effect) | 4/9 | NA | 1/2 | 24/31 | 7/16 |

(LSD) procedure). M1 activity was also significantly larger than that of A5 and PMd (p=0.002 and 0.0009 respectively). The perturbation response was not significantly different across other areas (p>0.07 in all other comparisons).

We explored how trial-by-trial changes in perturbation-related activity correlated with variations in the timing of the corrective response. Neural activity (50 to 100 ms post-perturbation) for each trial in the neuron's preferred torque direction was compared to the time of joint reversal (max distance from start position in joint space before returning to central target). We found the activity of 22/213 M1 neurons correlated with kinematic changes in the postural perturbation task (p<0.05 threshold). This low number partially reflects our study design in which only a small number of trials were collected per load condition (n=10). Across the M1 population, the median correlation was -0.11 (p=0.002 in comparison with 0 using a Wilcoxon signed rank test). This low, although significant correlation is not surprising given that correlations between proximal arm muscle activity and motor corrections during similar time epochs are in the -0.2 or -0.3 range (*Crevecoeur et al., 2013*). Perturbation-related activity in the other cortical regions was not significantly correlated with motor corrections although the data samples were smaller than that for M1 (median correlation coefficient, A5: -0.014, A2: -0.001, S1: -0.11, PMd: 0.03, p>0.35 in all areas in comparison with 0 using a Wilcoxon signed rank test).

*Figure 1E* displays population signals of the cells with a significant perturbation response for each cortical region. Perturbation onset time for each cortical area was calculated based on the time point when the population signal crossed 3SD above baseline activity (and remained above for at least 20 ms), highlighting that perturbation-related activity arrived quickly in all cortical regions (onset time, A5:21 ms, A2:18 ms, S1:17 ms, M1:22 ms, PMd:25 ms).

Comparisons of perturbation-related activity to baseline activity for calculating the onset time is not sensitive to sample size, but assumes variance remains constant throughout time (see Materials and methods). As an alternate method, we used a one-sample running t-test to compare when the population signal was different from baseline activity (1 ms steps). We identified the first

**Table 2.** Neural activity [mean ± SD] in each area, 50-100 ms post perturbation minus baseline, in response to the perturbation and across tasks. sp/s: spikes per second.

|  | A5 | A2 | S1 | M1 | PMd |
|---|---|---|---|---|---|
| Perturbation response | 30.3 ± 27.4 sp/s | 38.7 ± 26.1 sp/s | 68.5 ± 50.1 sp/s | 43.3 ± 27 sp/s | 26.9 ± 31.6 sp/s |
| Differential activity across areas (Movie) | 11.2 ± 12.8 sp/s | 5.6 ± 16.9 sp/s | 8.2 ± 20.8 sp/s | 14.4 ± 19 sp/s | 8.3 ± 18.3 sp/s |
| Change ratio across areas (Movie) | 36% | 15% | 9% | 36% | 40% |
| Differential activity across areas (IN/OUT) | -0.3 ± 6.3 sp/s | -2.2 ± 10.2 sp/s | -1.1 ± 6 sp/s | 7.4 ± 16.5 sp/s | 3.2 ± 14.2 sp/s |

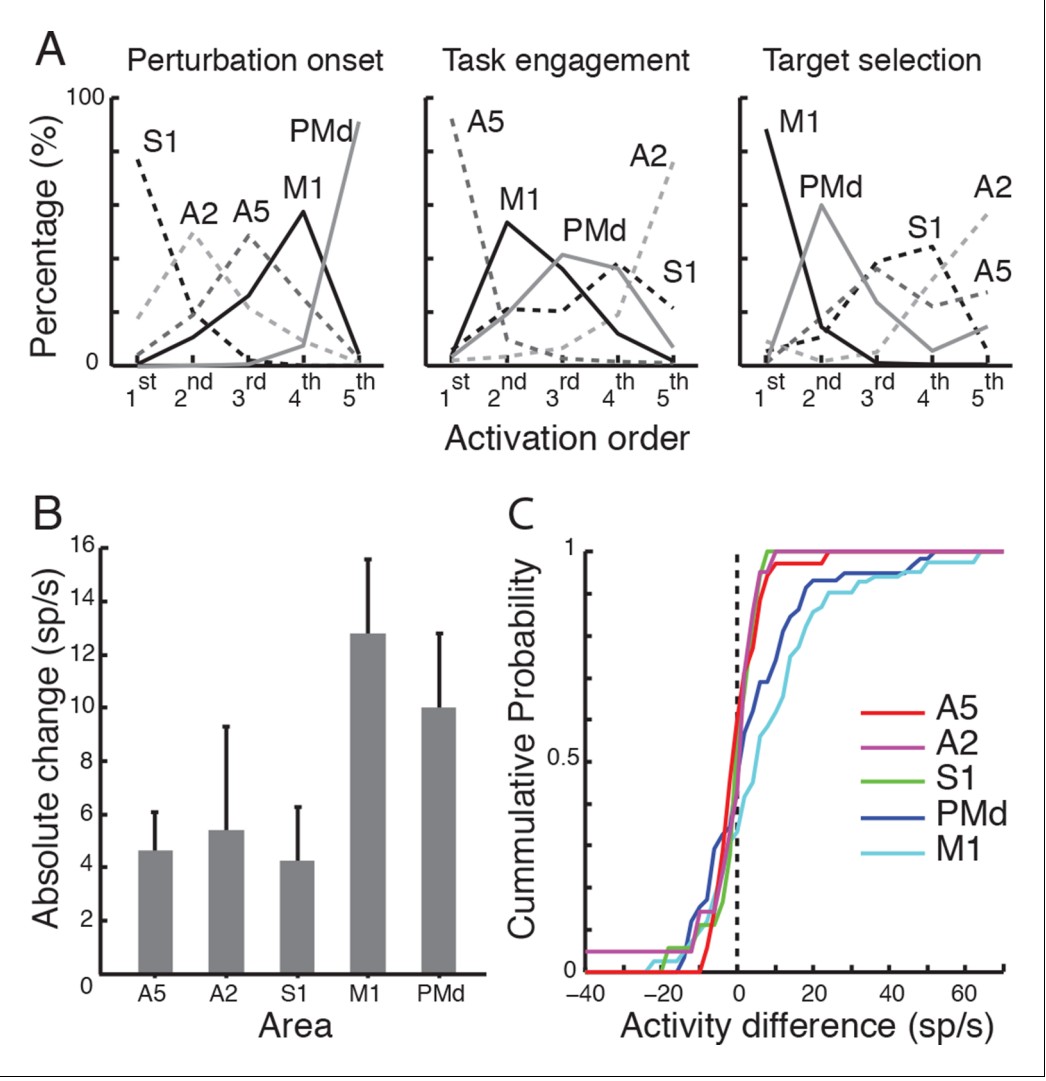

**Figure 2.** Response onset across tasks. (**A**) Different areas were ranked based on their activity/task onset times across 10000 random iterations of data in each area. Proportion of times each area assumed a rank is plotted for each task. (**B**) Absolute and (**C**) cumulative distributions of change in activity across targets (in the IN/OUT task) in each area.

point in time that the evoked population activity became significantly different from 0, and remained significant for at least 20 ms (*Figure 1E*, the first point in time the lower edge of the shaded area depicting 2SE rises above the baseline level). Response onset times, using this technique were generally similar to those observed using the 3SD technique (A5:25 ms, A2:25 ms, S1:19 ms, M1:24 ms, PMd:34 ms). Thus, sensory feedback from the limb is rapidly transmitted throughout sensory and motor cortical regions.

We used a bootstrap technique to identify the likely order of onset times for perturbation responses and task-dependency across cortical areas. To do so, we resampled (with replacement) cells in each population and then calculated the response onset and rank-order of each cortical area. We performed this procedure 10,000 times, and calculated the proportion of times each cortical area assumed a rank across the resampled data. The left panel in *Figure 2A* shows the percentage activity onset times in each area were ranked from 1st to 5th. This analysis revealed that primary somatosensory cortex was the first to respond (with S1 and A2 ranking 1st 95% of the time-S1:77%

& A2:18%), premotor cortex was the last to respond (ranking last, 92%), and A5 and M1 responded in between these extremes.

## Task engagement alters spatiotemporal pattern across sensory and motor cortex

We next examined whether perturbation responses were altered when the animal was not engaged in a limb motor task. Perturbation responses during the Posture Task were compared with those observed when the monkey was sitting quietly watching a movie and not required to respond to the perturbation (Movie Task, *Figure 1B*, *Omrani et al., 2014*). In the Movie Task, the monkey watched a movie displayed on the virtual reality display as the robot moved the monkey's unseen hand to the central target. At a random time point, a perturbation was applied to the limb (same load conditions as used in the Posture Task). Hand and joint motions were similar across tasks for the first 100 ms. Importantly, corrective movements, and corresponding long-latency muscle stretch responses in the Movie Task, were greatly diminished as compared to the Posture Task (average muscle response to the perturbation started 34 ms post-perturbation and differentiated across tasks 45 ms post-perturbation, *Omrani et al., 2014*).

Activity of 416 neurons was recorded in the Posture and Movie Tasks, with 305 displaying significant perturbation responses in the Posture Task (refer to *Table 1* for details on the number of neurons responsive to perturbation in each area). Perturbation-related activity was commonly modulated between the Posture and Movie Tasks (*Figure 3A*, refer to *Table 1* for details on the number of neurons). In most cases, the response was smaller in the Movie Task (refer to *Table 1* for details on the number of neurons).

The magnitude of the perturbation response was significantly different across tasks and areas (mixed ANOVA with task as within-subject and area as between-subject variables, $p<0.0001$, $F_{(1,300)}=62.7$ for the task effect and $p=0.04$, $F_{(4,300)}=2.5$ for the interaction of task and area). The magnitude of the reduction was significant in A5, M1 and PMd ($p<0.01$, one sample t-test), and marginally not significant in S1 and A2 ($p=0.086$ & $0.052$ respectively).

In order to evaluate the task effect directly, we compared the differential signal across tasks using a One-way ANOVA (with area as fixed variable and cells as random variables, $p=0.04$, $F_{(4,300)}=2.5$). Decreases in activity across tasks were only significant in A5, M1 and PMd (one sample t-test, $p<0.0001$ in A5 and M1, $p=0.0016$ in PMd and $p=0.052$ & $0.086$ in A2 and S1 respectively, refer to *Table 2* for details on the differential activity in each area across tasks). A post-hoc analysis (LSD) revealed that M1 differential response was significantly bigger than the differential response in A2 ($p=0.009$) and PMd ($p=0.03$). No other significant differences were found among other areas ($p>0.13$ in all other comparisons).

We also quantified the relative change in the perturbation response in each area (i.e. evoked response in the Posture minus that in the Movie tasks, divided by the evoked response in the Posture Task, refer to *Table 2* for details on the relative change in each area across tasks) and found it to be significantly different across areas (One way ANOVA with area as fixed variable and cells as random variables, $p=0.019$, $F_{(4,300)}=3$). A post-hoc analysis (LSD) revealed that the relative changes in the perturbation response were significantly smaller in S1 and A2 than the change ratio in other areas (relative to A2, A5:$p=0.036$, M1:$0.024$ and PMd:$0.016$, and relative to S1, A5: $p=0.025$, M1:$0.019$, PMd:$0.012$). The change ratio was not significantly different between A5, M1 and PMd (relative to A5, M1:$p=0.95$, PMd:$0.65$, also $p=0.57$ in M1-PMd comparison) or between A2 and S1 ($p=0.64$).

Of particular importance is that the timing of the change in the perturbation response varied across the cortical areas. The population signal for A5 was reduced in the Movie as compared to the Posture Task at 23 ms, effectively at the same time as the onset of the initial perturbation response in this cortical region (*Figure 3B,C*, differential signal >3SD of baseline). In contrast, other cortical areas only displayed a significant reduction in the population signal at ∼40 ms or later after the applied load (M1: 38 ms, PMd: 42 ms, S1: 40 ms and A2: 70 ms).

The running t-test was also used to identify when the population activity was different between tasks. In the Posture-Movie tasks comparison, we identified the first point in time that the differential signal (activity in the Posture Task – activity in the Movie Task) became significantly different from 0, and remained significant for at least 20 ms (*Figure 3C*, the first point in time the lower edge of the shaded area depicting 2SE rises above baseline). Response differentiation times were 25 ms, 45 ms

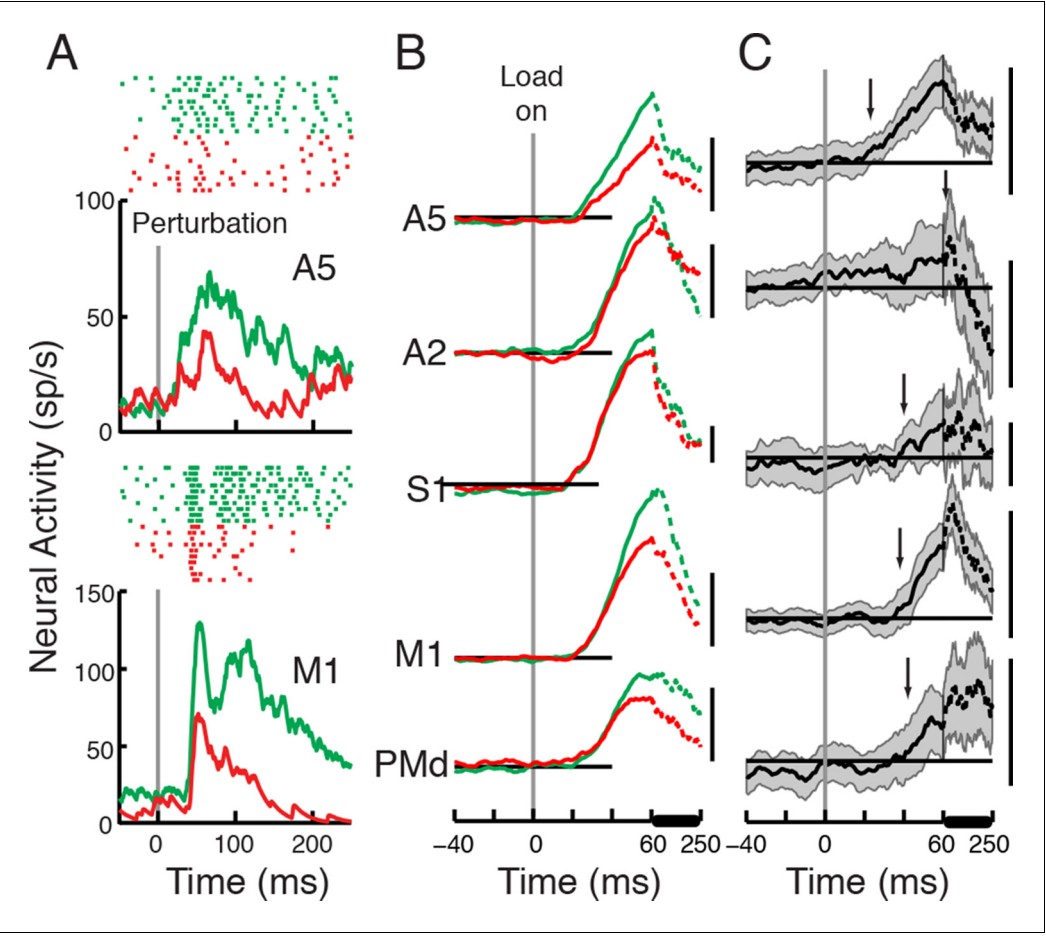

**Figure 3.** Perturbation responses compared across the Posture and Movie Tasks. (**A**) Perturbation response across the Posture (green) and Movie (red) tasks (same neurons as in *Figure 1D*). (**B**) Population signal and (**C**) differential signal across tasks (Posture - Movie) in each cortical area. Each scale bar represents 20 sp/s.

and 52 ms for A5, M1 and PMd, respectively. As shown in *Figure 3C*, the 2SE shaded area never rises (and stays) above zero in S1 and A2, hence no response differentiation time was detected for these areas. The failure to identify these onset times likely reflects the influence of sample size on the running t-test.

Finally, our bootstrap analysis identified the most likely order of response differentiation to be A5 (ranking 1st 89%, all other regions each less than 5%, middle panel *Figure 2A*), then M1, PMd, S1 and last A2 (ranking last 73%).

## Task selection alters spatiotemporal pattern across sensory and motor cortex

Somatosensory feedback also permits rapid transition from one motor task to another (*Hammond et al., 1956*; *Rothwell et al., 1980*; *Johansson and Flanagan, 2009*). Our last experiment quantified how perturbation responses were altered across sensory and motor cortices when the load instructed the monkey to move to a second spatial target. Loads in this target selection task either pushed the hand into the spatial target (IN) or away from it (OUT), eliciting a larger corrective response in the latter condition(*Figure 1C*, *Pruszynski et al., 2014*). Note that in this condition, the monkey should always be engaged in the task but produce different magnitudes of response for each target. Differences in muscle responses between the IN and OUT targets begin ~85 ms after the applied load (See Materials and methods). The onset of this differential response is

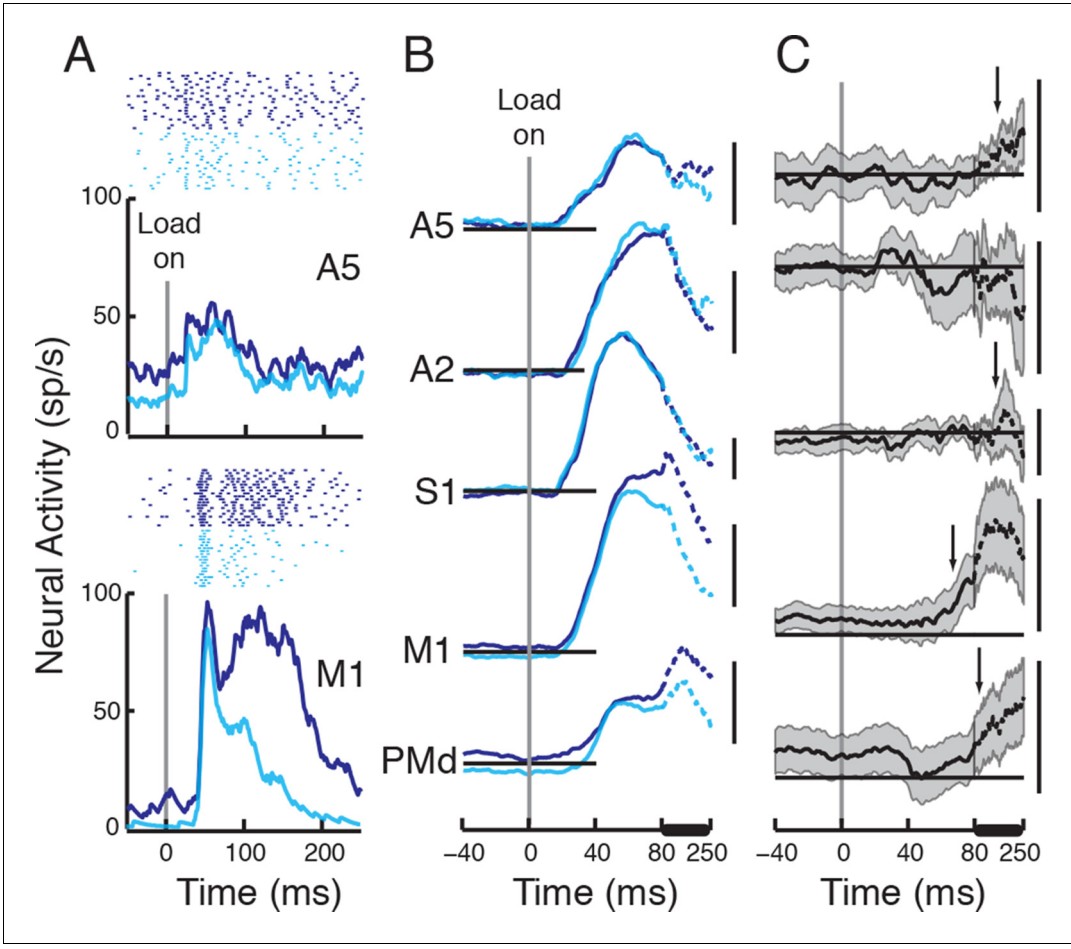

**Figure 4.** Perturbation responses compared across different target positions. (**A**) Perturbation response for the OUT (navy blue) versus IN (cyan) targets (same neurons as in *Figure 1D*). (**B**) Population signal and (**C**) average differential signal across tasks (OUT-IN) in each cortical area. Each scale bar represents 20 sp/s.

slower than our previous study (~70 ms, *Pruszynski et al., 2014*), which likely reflects that that study used a background load to prime the muscles, whereas the present study did not.

Activity of 263 neurons was recorded in the IN and OUT tasks; with 215 displaying significant perturbation responses in the OUT target Task (refer to *Table 1* for details on the number of neurons responsive to perturbation in each area). Perturbation responses in M1 and PMd were commonly altered between the IN and OUT targets (*Figure 4*, refer to *Table 1* for details on the number of neurons). Altered responses were also observed in A5, but rarely in somatosensory cortex. Significant increases in activity in the OUT target were predominantly observed in M1, but not in A5 and PMd (refer to *Table 1* for details on the number of neurons).

The perturbation magnitude was not significantly different across targets but was significantly different across areas (mixed ANOVA with target as within-subject and area as between-subject variables, p=0.2, $F_{(1,211)}=1.64$ for the target effect and p=0.005, $F_{(4,211)}=3.85$ for the interaction of target and area). We also directly compared the differential signal across targets (OUT-IN 50–100 ms post perturbation) using a one-way ANOVA (with area as the fixed variable and cells as random variables, p=0.005, $F_{(4,211)}=3.85$). M1 was the only area, which had a significant difference in its activity across the two targets in the 50–100 ms period (one sample t-test, p=0.0001 in M1, p=0.08 in PMd and p>0.3 in A5, A2 and S1, refer to *Table 2* for details on the differential activity in each area across tasks). A post-hoc analysis (LSD) revealed that M1 differential response between IN and OUT targets was significantly larger than in A5 (p=0.005), A2 (p=0.004) and S1 (p=0.017) but not different than

that of PMd (p=0.08). The differential activity was not significantly different across other areas (p>0.1 in all other comparisons).

We examined whether the absence of any significant target effect in A5 and PMd was due to the fact that half the cells were increasing and half were decreasing their activity between IN and OUT targets (see *Pruszynski et al., 2014*). We therefore compared the absolute change in activity across targets (*Figure 2B*, absolute activity change in A5:4.6 sp/s ± 4.2 sp/s, A2:5.4 sp/s ± 8.9 sp/s, S1:4.2 sp/s ± 4.3 sp/s, M1:12.8 sp/s ± 12.8 sp/s, PMd:10 sp/s ± 10.5 sp/s). The absolute change in magnitude was significantly different across areas (One way ANOVA, p<0.001, $F_{(4,211)}$=5.92). A post-hoc analysis (LSD) revealed that the absolute change in activity in M1 was significantly larger than in A5 (p<0.001), A2 (p=0.004) and S1 (p=0.002) but not different than in PMd (p=0.13). The absolute change in activity was also bigger in PMd compared to the absolute change in A5 (p=0.016) and S1 (p=0.04). We also examined the cumulative distributions of change in activity for each area (*Figure 2C*). A two-sample Kolmogorov-Smirnov test revealed that the distribution was significantly different in M1 compared to A5 (p<0.001), A2 (p=0.001), S1 (p=0.004), and in PMd compared to A5 (p=0.05). All other comparisons were not significant (p>0.1 across all areas).

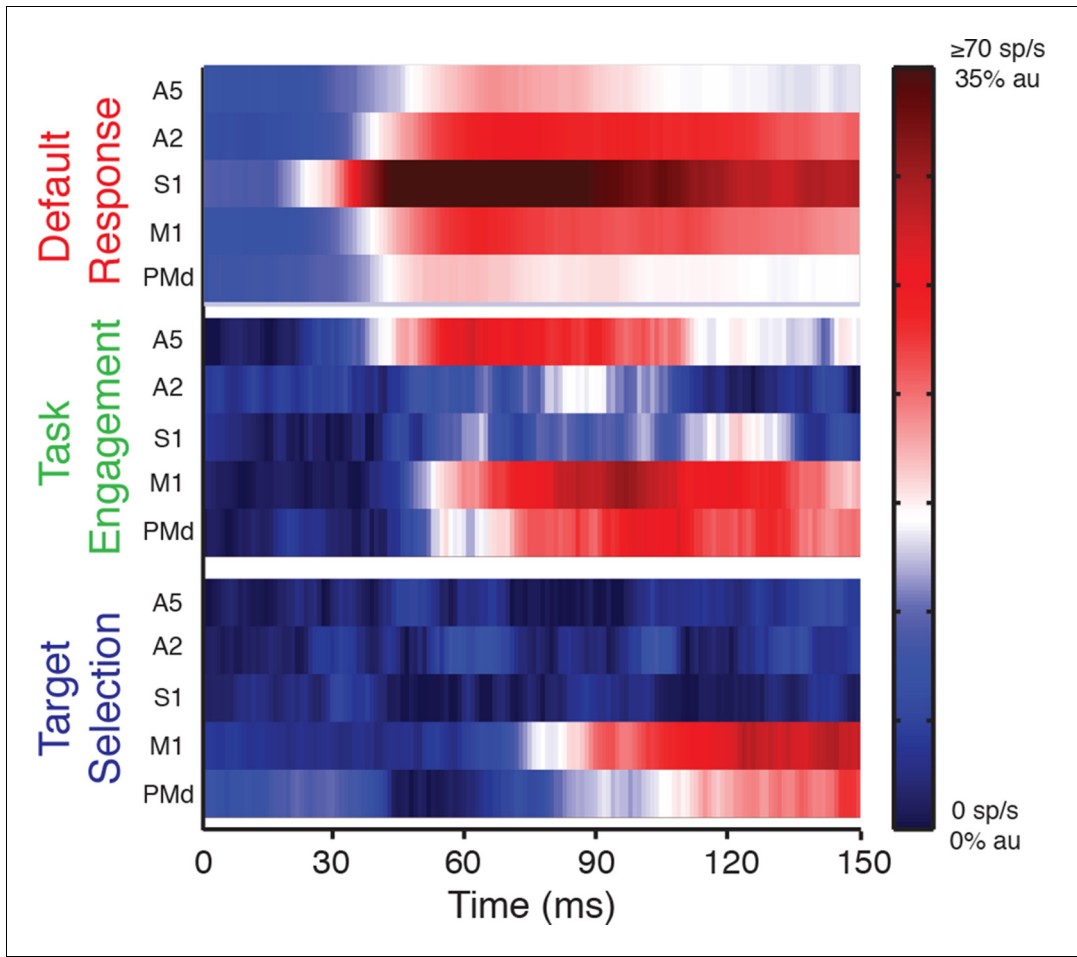

**Figure 5.** Context-dependent patterns across sensorimotor cortex - Default response (top panel), is represented by activity patterns in the Movie Task. Task engagement (middle panel), is represented by the differential signal between the Movie and Posture Tasks. Target selection (bottom panel), is represented by differential signal between the OUT and IN targets. Activity is plotted using a color map. In the default response, population response is capped at 70 sp/s. Differential signals are normalized to their maximum response in the Posture Task (au).

The timing of the change in the perturbation-response for the IN and OUT targets also varied across cortical areas. Differences in the population signals for the IN and OUT targets were first observed in M1, 66 ms after perturbation onset, and then in PMd at 98 ms (*Figure 4B,C*, differential signal >3SD of baseline). In contrast, population signals in S1 and A5 did not show any difference across targets until ~150 ms post-perturbation, and the differential signal in A2 never passed threshold in this task.

For the IN-OUT target comparison, the baseline activity was already significantly different in M1 & PMd (M1:2.4 sp/s ± 7.2 sp/s, PMd:3.6 sp/s ± 12.5 sp/s, one sample t-test, p=0.009 & 0.029 in M1 & PMd respectively). Thus, in using the running t-test technique, we compared the differential activity relative to its baseline difference rather than 0. We identified the first point in time the differential activity was significantly different from baseline, and remained significant for at least 20 ms (*Figure 4C*, the first point in time the lower edge of the shaded area depicting 2SE rises above the baseline line). With this technique, response differentiation times were 72 ms, 180 ms and 160 ms for M1, PMd and A5 respectively. As observed (*Figure 4C*), the 2SE shaded area never rises (and stays) above the zero line in S1 and A2, and hence no response differentiation time was detected for these areas.

The fact that neurons could increase or decrease activity between the IN versus OUT target could impact the onset time for observing differences in the population signals associated with each target (See also *Heming et al., 2016*). To rule this out, we reversed the differential sign for cells that significantly decreased their activity in the OUT versus IN target when calculating the difference in the population signals (this means we used IN-OUT in these cells rather than OUT-IN). Differences in the population signals tended to be slightly earlier (3SD technique, M1:58 ms, PMd:78 ms, A5:157 ms, S1:173 ms and no differentiation time for A2), but the order of onset times remained the same with only M1 and PMd showing significant onset times before 100 ms.

Finally, the bootstrap technique identified the most likely order of response differentiation to be M1 first (86%, all others regions each less than 5%, except for A2 which was 8%, right panel in *Figure 2A*), PMd second (58%), followed by S1 and A5 (similar ratio of 35%) and finally A2 (55%).

*Figure 5* provides an overview of the main results on how perturbation-related activity is transmitted across sensory and motor cortical regions, and how this spatiotemporal pattern of activity is altered by behavioral context. The top panel highlights perturbation responses when the monkey is not rewarded for responding to the mechanical load, termed the 'default response'. In this case, limb feedback is rapidly transmitted across the cortex with the greatest and earliest activity in S1 quickly followed by responses in adjacent cortical regions. The middle panel displays how the perturbation response changes between the Movie and Posture Tasks, termed 'task engagement'. In this case, the perturbation response increases immediately in A5 and then in motor cortical regions with minimal effect in primary somatosensory cortex. Finally, the bottom panel displays how perturbation responses are altered when they cue the generation of a goal-directed movement, termed 'target selection'. In this case, perturbation-responses first increase in M1, and then PMd, with minimal effect in sensory areas until 150 ms.

## Discussion

The neural basis of feedback processing for voluntary motor actions has not received much attention for the last 30 years. At that time, the prevailing view was that task-dependent changes in long-latency motor responses, that occur 50 to 100 ms after a mechanical disturbance, were provided by a transcortical feedback pathway from primary somatosensory to primary motor cortex (*Desmedt, 1977*; *Brooks, 1986*). The timing of mechanical responses in M1 are sufficiently fast to generate these long-latency responses. Classic work by Evarts and Tanji illustrated that perturbation responses in M1 could be modulated when the disturbance was the cue for the monkey to push or pull a lever (*Evarts and Tanji, 1974*). Further studies implicated the dentate nucleus in this task-dependent feedback processing (*Meyer-Lohmann et al., 1975*; *Vilis et al., 1976*; *Strick, 1983*; *Hore and Vilis, 1984*), likely through its connections to M1 (*Orioli and Strick, 1989*; *Dum and Strick, 2003*). Based on this evidence, it has been generally assumed that there was minimal overlap in cortical circuits involved in online feedback processing from the limb (S1 and M1), and the broad parietal and frontal cortical circuits involved in motor planning and initiation.

The present study provides the first examination of the relative timing and magnitude of perturbation-related activity across sensory and motor cortices in non-human primates. Here we show that even when the animal is not engaged in a limb motor task, limb afferent feedback is still rapidly transmitted to many sensory and motor cortical regions, beginning in S1 and then spreading to adjacent cortical regions (*Figure 5*). The magnitude of the perturbation response is also greatest in S1 and diminishes spatially across the cortex. This gradient in the timing and magnitude of perturbation-related activity across the cortical surface may be generated by intra-cortical communication, although subcortical sensory feedback pathways may also be involved (*Cappe et al., 2007*; *Padberg et al., 2009*).

While muscle spindle afferents are assumed to play a dominant role for eliciting the perturbation-related activity observed in the present study (*Lucier et al., 1975*), cutaneous afferents likely also contribute, particularly in S1, signaling skin stretch caused by the perturbation. However, the very fastest responses are likely due to muscle afferents even in area 3b (*Heath et al., 1976*). Neurons in S1 with cutaneous receptive fields are broadly tuned to the direction of movement during reaching (*Cohen et al., 1994*; *Prud'homme et al., 1994*), much like neurons in M1 with receptive fields from shoulder and elbow muscles (*Scott and Kalaska, 1997*). Correspondingly, we expect both muscle and cutaneous afferents will be broadly tuned to loads applied to the shoulder and elbow, and contribute to feedback responses observed in the present study. Yet the differential physiological contributions of each of these feedback sources to control are an interesting question, warranting further investigation.

M1 was the first cortical region to display changes in perturbation-related activity when the applied load was used to initiate movements to the OUT versus the IN target, and occurred prior to changes in proximal limb muscle activity. This suggests M1 is the primary cortical source for implementing this aspect of control (along with dentate). While changes in perturbation-related activity was only observed in PMd at the same time or later than that observed in limb muscles, this cortical area did show changes in baseline activity for the IN and OUT targets before perturbation onset. One can also see a small (non-significant) dip in the differential signal in *Figure 4C* at about 40 ms post-perturbation. Thus, PMd may play some role in this process of using sensory feedback for action selection. In contrast, all parietal regions examined in the present study displayed no change in perturbation-related activity until ∼150 ms when differences in limb motion begin to emerge (*Pruszynski et al., 2014*), suggesting none of these regions were involved in this aspect of control. However, a more thorough examination of area 3a, that has direct projections onto motoneurons of limb muscles, is warranted (*Rathelot and Strick, 2006*).

Of particular note is that engagement of the limb in a motor task increases the perturbation response immediately in A5 (∼25 ms). In contrast, the initial response was unaltered in other cortical regions, but they then increased 15 ms later or more, potentially driven by A5. Parietal area 5 has been implicated in somatomotor and visuomotor processing for limb motor actions (*Mountcastle et al., 1975*; *Chapman et al., 1984*; *Kalaska, 1996*; *Debowy et al., 2001*; *Reichenbach et al., 2014*). Preparatory activity in this cortical region reflects the selective use of the limb for a subsequent motor action (*Cui and Andersen, 2011*). Our data suggests a potential role of A5 in online control, consistent with recent studies demonstrating TMS over medial intraparietal sulcus in humans (approximate analog to A5 in NHPs) prolongs corrective responses to mechanical disturbances during reaching (*Reichenbach et al., 2014*). Thus, posterior parietal cortex is not only important for online control of visual feedback (*Desmurget and Grafton, 2000*), but also somatosensory feedback. The fact that perturbation-related activity was altered between the movie and postural tasks initially suggests a role in the control policy rather than state estimation. However, changes in perturbation-related activity in A5 were related to whether the animal was engaged or not in a behavioral task. There was no change in its response when using sensory feedback to select a new goal, as the animal remained engaged in a task before and after this selection. Thus, it may instead play a role in linking state estimation to the control policy for online control.

The present study explored how task engagement and target selection alters the spatiotemporal pattern of perturbation-related activity across sensory and motor cortices. Our results suggest that multiple areas are activated in response to sensory feedback, and activity in each area reflects processing of different aspects of the task. This concurrent processing of information across different areas could cumulatively shape the observed output generated in each task (*Ledberg et al., 2007*; *Cisek and Kalaska, 2010*; *Siegel et al., 2015*).

Task-dependent changes in perturbation-related activity in cortex preceded corresponding changes in muscle responses, and thus, these cortical circuits are the likely source for these task-dependent changes in motor output. However, long-latency motor responses are extremely complex (see *Shemmell et al., 2010*; *Pruszynski and Scott, 2012* for reviews) and consider many factors such as limb mechanics (*Kurtzer et al., 2008*; *2009*), goal-directed corrections associated with the shape of spatial goal (*Nashed et al., 2012*), presence of obstacles in the environment, and selection of alternate goals (*Nashed et al., 2014*). We predict that distributed frontoparietal circuits (and cerebellum) also provide the neural substrate to generate these other complex corrective responses, a hallmark of highly skilled motor behaviors (*Scott, 2012*).

## Materials and methods

### Subjects and apparatus

Three male non-human primates (*Macaca mulatta*, 10–17 kg) were trained to perform whole limb visuomotor tasks while attached to an exoskeleton robot (KINARM, BKIN Technologies, Kingston, Ontario, Canada). The robot permitted combined flexion and extension movements of the shoulder and elbow in the horizontal plane and applied loads to the shoulder and/or the elbow independently. Two monkeys (Monkey X & A) used a right-arm robot and one monkey (Monkey P) used a left-arm robot. Targets and hand visual feedback were presented to the monkeys, in the horizontal plane, using an overhead monitor and a semitransparent mirror. Hand position was represented by a white circle (5 mm diameter) positioned at the tip of the index finger. The Queen's University Animal Care Committee approved all experimental procedures (Protocol 1348).

### Behavioral tasks

Throughout the experiment different combinations of shoulder and/or elbow torques were applied to the monkey's arm. Three tasks were performed and varied in how the monkeys were required to respond to the perturbations. In the Posture Task (*Herter et al., 2007*), the monkey was instructed to maintain its hand at a central target (visual: 12 mm diameter, acceptable window: 16 mm diameter). At a random time (1000–1500 ms), the limb was perturbed with one of nine combinations of loads applied to the shoulder and/or elbow (flexor, extensor or null), and the monkey had to return its hand to the target within 750 ms of the perturbation time (*Figure 1A*). Each perturbation lasted 300ms and the size of the load varied with the size of the monkey (Monkey P & X, 0.24 Nm and monkey A, 0.32 Nm). Load magnitudes were adjusted in the bi-articular load directions to compensate for larger hand motions induced in these directions (*Herter et al., 2007*). Each load combination was presented randomly in a block of trials and the monkey was required to complete 10 blocks in a set.

In the Movie Task (*Omrani et al., 2014*), monkeys were not required to do anything in response to the perturbation. Task-related visual feedback (i.e. target position and hand position) was replaced by a movie and the monkeys were trained to quietly watch the movie. The robot moved the hand to the central target at the beginning of each trial. The hand was then perturbed using the same 9 load combinations as in the Posture Task (*Figure 1B*). The monkey was rewarded irrespective of its response to the perturbation.

In the IN/OUT task (*Pruszynski et al., 2014*), the monkey started each trial by maintaining its hand at a central target (12 mm diameter), and moved its hand to a second target (2.5 cm radius) following a perturbation (the load remained on for 1500 ms). The perturbation was the load combination from the Posture Task that elicited the largest response in the neuron/muscle presently being recorded. The location of the second target was strategically chosen such that the load either pushed the hand in (IN), or away (OUT) from it (*Figure 1C*; location of the second target could also remain aligned with the initial central target, but data not analyzed in this study). The monkey had to move to the second target within 750 ms and remain there for an additional 1 s. IN and OUT target trials were randomly interleaved and 20 repeat trials were recorded for each target.

Normally an experimental session was composed of a fixed order of tasks: first the Posture Task, then the Movie Task, followed by another repeat of the Posture Task and finally the IN/OUT task. A reduced version of the experiment was performed near the end of the recording session; in which one set of the Posture and the Movie Tasks were randomly presented followed by one set of the IN/

OUT task. In recording sessions from S1, receptive field properties of the neurons were investigated following the last experimental block.

## Data collection

Neural data was recorded from shoulder/elbow regions of the primary somatosensory areas 1&3 (S1), primary somatosensory area 2 (A2), parietal area 5 (A5), primary motor cortex (M1) and dorsal premotor cortex (PMd), using standard extracellular recording techniques (*Herter et al., 2007*; *Omrani et al., 2014*; *Pruszynski et al., 2014*). The neural data was initially sorted online for single units (Plexon Inc., Dallas), then confirmed and examined further offline using the Plexon offline sorter.

Recording chamber and penetration sites were chosen using monkey atlas coordinates (*Paxinos et al., 2008*) and also MRI imaging (for Monkeys X & A). Single tungsten microelectrodes (FHC, Bowdoin) were advanced in the cortex until neural activity was recorded. We verified the location of shoulder and elbow areas of M1 by eliciting muscle twitches using microstimulation (*Stoney et al., 1968*) and the response of neurons to passive movement of the joints. The shoulder/elbow regions in other cortical areas were generally at the same laterality as that in M1. We tested sensory receptive fields of neurons particularly in the first few sessions of recording in a new cortical area to make sure we were in the shoulder/elbow representation. While recording in the sensory cortices, we also tried to differentiate the best modality of stimulus for each neuron (e.g. cutaneous, soft touch, deep touch or joint movement). However, the robotic device attached to the arm made it difficult to dissociate whether sensory responses during manual examination were related to muscle or cutaneous afferents. The majority (~85%) of our recordings from A5 were performed over the convexity of the cortex (area 5d). Neural recordings in S1 were equally on the surface (putative Area 1), bank of post-central sulcus (putative area 3b) and deep in post-central sulcus (putative area 3a).

In one monkey (P), we verified our recording areas post-mortem. We used Paraformaldehyde to perfuse the monkey and its brain. Right before removing the chamber from the skull, we inserted several pins to known coordinates within the chamber. We then photographed the brain and sketched the location of the sulci (*Figure 1—figure supplement 1*). Post mortem penetration locations have yet to be performed in the other monkeys.

We also recorded electromyographic (EMG) activity of proximal arm muscles during the tasks. The EMG recordings were scored from 1 to 5 (based on recording quality, gain of the signal, signal-to-noise ratio, and whether the muscle looked active in the task). Muscles that scored 3 and higher were included in our analysis. EMG signals were band-pass filtered (10–150 Hz, two-pass, third-order Butterworth) and full-wave rectified. Each trial was aligned based on the perturbation onset. The EMG data related to posture and movie tasks were presented previously (*Omrani et al., 2014*). For the IN/OUT task, we recorded EMG from 3 to 6 proximal limb muscles in 9 sessions in one monkey (Monkey P). Nineteen samples (representing all the major muscles involved in flexion and extension of the shoulder and elbow joints) were identified as good quality (score 3 or higher on subjective rating scale out of 5) and had significant perturbation responses ($p < 0.05$).

## Data analysis

Spike times were extracted from the Plexon files into Matlab (Mathworks, Natick). Spike-density functions were generated by convolving spike time-stamps with asymmetric double-exponential kernels (1 ms rise- and 20 ms fall-time, [*Thompson et al., 1996*]). We consider cell activity 50–100 ms post-perturbation corresponding roughly to the long-latency epoch for muscle activity. The load combination with the largest response (50 to 100 ms post-perturbation) was then selected as the neuron's preferred-torque direction (PTD). The neuron's activity in its PTD was then compared to its activity in the null load condition (catch trial) using a two-sample t-test. If the comparison was significant, the activity of the cell in its PTD was used for further analyses (SPSS, IBM, New York).

Single cell activities in each area were averaged to calculate the perturbation population response for each area. We determined the first point in time that the activity of a cell/muscle/population passed a defined threshold (baseline + 3 SD of baseline activity) and remained above this threshold for at least 20ms (to avoid capturing random transient responses). The baseline period consisted of cell/population activity 100 ms prior to the perturbation (average population baseline for each area is represented as insets in *Figure 1E*). A similar approach was used to identify when population

signals associated with different conditions (Posture versus Movie Tasks, or IN and OUT targets) were significantly different (differential signals). We compared the onset times of perturbation responses and differential signals across cortical areas using a bootstrap technique, resampling (with replacement) cells in each population 10000 times, and then calculating the response onset for each iteration. We also rank ordered the onset across different areas in each iteration and calculated the percentage of times activity in one area preceded that of others. In calculating the percentages, we also included iterations where the population or difference signal did not pass 3SD.

## Acknowledgements

This work was supported by the Canadian Institutes of Health Research (CIHR). MO received a Vanier Doctoral Award from CIHR. JAP received salary awards from CIHR and the Human Frontier Science Program. SHS is supported by a GSK-CIHR Chair in Neuroscience. We thank Kim Moore, Simone Appaqaq and Justin Peterson for their technical support and Drs. Doug Munoz, Gunnar Blohm, DJ Cook, Roozbeh Kiani, Baktash Babdi and the Scott lab for their comments.

## Additional information

### Competing interests

SHS: Co-Founder and Chief Scientific Officer of BKIN Technologies that commercializes the robotic technology used in this study. The other authors declare that no competing interests exist.

### Funding

| Funder | Author |
| --- | --- |
| Canadian Institutes of Health Research | Mohsen Omrani<br>Chantelle D Murnaghan<br>J Andrew Pruszynski<br>Stephen H Scott |

The funders had no role in study design, data collection and interpretation, or the decision to submit the work for publication.

### Author contributions

MO, Performed experiments, Analyzed data, Interpreted results of experiments, Prepared figures, Drafted manuscript, Edited and revised manuscript, Approved final version of manuscript; CDM, Performed experiments, Edited and revised manuscript, Approved final version of manuscript; JAP, Conception and design of the research, Performed experiments, Edited and revised manuscript, Approved final version of manuscript; SHS, Conception and design of the research, Interpreted results of experiments, Drafted Manuscript, Edited and revised manuscript, Approved final version of manuscript

### Author ORCIDs

J Andrew Pruszynski, http://orcid.org/0000-0003-0786-0081
Stephen H Scott, http://orcid.org/0000-0002-8821-1843

### Ethics

Animal experimentation: The Queen's University Animal Care Committee approved all experimental procedures. (Protocol 1348)

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
