## [Decision Letter]

Thank you for submitting your work entitled "Distributed task-specific processing
of somatosensory feedback for voluntary motor control" for consideration by
*eLife*. Your article has been reviewed by two peer reviewers, and the
evaluation has been overseen by a Reviewing Editor and David Van Essen as the Senior
Editor.

The reviewers have discussed the reviews with one another and the Reviewing Editor has
drafted this decision to help you prepare a revised submission.

The authors should highlight the novelty of the findings in relation to previous
studies, as there was much debate about the advances that this study provides. The
authors should also address the following points:

1) Trial-by-trial analysis correlation between the timing of the corrective responses
and the timing of neuronal activity.

2) Determination if the cells had receptive fields on the shoulder/elbow.

3) Discussion of how the activation of different receptor types would influence the
responses evoked in different areas.

4) Influence of early- versus long-latency responses in the conclusions of the
study.

---

## [Author Response]

*The authors should highlight the novelty of the findings in relation to previous
studies, as there was much debate about the advances that this study
provides.*

Our Introduction was perhaps too subtle with regards to articulating the novelty of our
approach. We have re-written the Introduction to provide:

1) A more theory driven approach based on Optimal control and highlight the dichotomy of
state estimation and control policy;

2) The value of using mechanical disturbances to probe voluntary control, and;

3) The use of multiple behaviors to identify if and how a brain region reflects
different aspects of control.

We think this provides a more transparent presentation of our approach. We have also
modified the Discussion a bit to align with the changes in the Introduction. While the
Introduction now uses optimal control principles to introduce the motivation for the
study, we believe the results are not necessarily bound within this framework.

*The authors should also address the following points:*

*1) Trial-by-trial analysis correlation between the timing of the corrective
responses and the timing of neuronal activity.*

This is an interesting analysis. Unfortunately, we did not consider this type of
analysis when designing our experiment as we only have 10 trials per load condition
making it difficult for individual neurons to reach statistical significance. At the
single cell level, there are some neurons that had significant correlations with joint
reversal times (10% in M1). At the population level, M1 was the only cortical region
that displayed a significant correlation. The low correlation level for the M1
population (r= -0.11) is not surprising as even arm muscle activity has rather low
correlations (Crevecoeur et al., 2013). We have added this analysis to the Results
section.

*2) Determination if the cells had receptive fields on the
shoulder/elbow.*

We specifically targeted our recordings in areas of the cortex that correspond with
shoulder/elbow movement or have sensory responses to shoulder/elbow manipulations (i.e.
cutaneous or proprioceptive response). These areas were recognized in mapping sessions,
which were performed following the recording chamber implantation and before behavioral
recording sessions were performed. In these sessions, as a general rule, we first
identified M1 and then explored other rostral-caudal areas at this same lateral distance
from the midline. If the cells had responses to hand and wrist movement, we then moved
our penetrations to more lateral sites to find shoulder/elbow cells. We also tested cell
receptive fields more closely in the first few sessions of recording in a new area to
make sure we were in the shoulder/elbow region for a given cortical area. We now explain
this process in more detail in the Methods section. We have also included the
penetration map for the monkey that was euthanized and in which we can relate cortical
surface maps to the penetration sites (refer to supplemental material).

*3) Discussion of how the activation of different receptor types would influence
the responses evoked in different areas.*

It is very difficult to differentiate the potential contribution of different receptor
types. Reaching studies highlight that neurons with cutaneous receptive fields in area
3b are broadly tuned to the direction of hand motion like all other cortical areas,
probably reflecting skin stretch in response to the perturbation (Cohen et al., 1994).
Both cutaneous and muscle afferents will likely generate similar patterns of broad
tuning to loads applied to the shoulder and elbow. Further, area 3b that is commonly
assumed to be receive purely cutaneous information actually receives substantive input
from muscle afferents (Heath et al., 1976). Muscle afferents tend to be a bit faster
than cutaneous although the difference can be small (Heath et al., 1976). We have added
a paragraph in the Discussion to raise the point that while muscle afferents likely play
a dominant role providing feedback of limb motion from the applied loads, cutaneous
afferents also likely contribute. We have also tried to explain why differentiating
these modalities is not easy while the animal is in the robot. We have also tried to
clarify in more detail the penetration sites within S1 in the Methods section.

*4) Influence of early- versus long-latency responses in the conclusions of the
study.*

R1 (short) and R2 (long) muscle responses have clear differences, which we have
articulated many times (and many others have as well). Notably, R1 responses are
generated at the spinal level and are fairly fixed responses that do not reflect
task-level corrective responses. In contrast, R2 responses are influenced by many
different behavioral level factors and occur at a time point when cortical activity can
contribute. These observations lay the groundwork for the present work to begin to
unravel how cortex may be involved in these goal-directed motor corrections that begin
in the R2 epoch. The comparisons across tasks allow us to identify when
perturbation-related activity is task-independent (like R1) and task-dependent (like
R2). We assume the issue is that we don’t link our task-dependent effects in cortex to
the motor periphery. Thus, we have added some statements in the Discussion to more
explicitly state that task-dependent changes observed in M1 for the IN/OUT task are
appropriate for driving the subsequent R2 muscle responses.